# ZERO REDUNDANCY DISTRIBUTED LEARNING WITH DIFFERENTIAL PRIVACY

## ABSTRACT

Deep learning using large models have achieved great success in a wide range of domains. However, training these models on billions of parameters is very challenging in terms of the training speed, memory cost, and communication efficiency, especially under the privacy-preserving regime with differential privacy (DP). On the one hand, DP optimization has comparable efficiency to the standard non-private optimization on a single GPU, but on multiple GPUs, existing DP distributed learning (such as pipeline parallel) has suffered from significantly worse efficiency. On the other hand, the Zero Redundancy Optimizer (ZeRO) is a state-of-the-art solution to the standard distributed learning, exhibiting excellent training efficiency on large models, but to work compatibly with DP is technically complicated. In this work, we develop a new systematic solution, DP-ZeRO, (I) to scale up the trainable DP model size, e.g. to GPT-100B, (II) to obtain the same computation and communication efficiency as the standard ZeRO, and (III) to enable mixed-precision DP training. Our DP-ZeRO, like the standard ZeRO, has the potential to train models with arbitrary size and is evaluated on the world's largest DP models in terms of the number of trainable parameters.

## 1 INTRODUCTION

Recent advances in differentially private (DP) deep learning have witnessed the power of large pre-trained models, achieving comparable accuracy to state-of-the-art (SOTA) non-private models across computer vision De et al. (2022); Bu et al. (2022a); Mehta et al. (2022); Xie et al. (2018), natural language processing Yu et al. (2021); Li et al. (2021); Bu et al. (2023a), and many other tasks. Similar to their non-DP counter-parts, it has been observed that larger DP models tend to have better performance. For example, the DP accuracy increases from 83% using RoBERTa-base (123M parameters) to 86% using RoBERTa-large (354M parameters) on GLUE datasets Li et al. (2021); Bu et al. (2023a); Yu et al. (2021); the DP BLEU score increases from 61 using GPT2-small (124M parameters) to 64 using GPT2-large (800M parameters) on E2E dataset Li et al. (2021); Bu et al. (2023a); a similar trend is also observed using ViT (Base/Large/Huge) up to 600M parameters to achieve state-of-the-art DP accuracy on ImageNet, around 81% at $\epsilon = 8$ Mehta et al. (2022).

Driven by this success and the surge of computational power, it is high time to enable DP deep learning at the same scale of the standard non-DP one, e.g., GPT3-175B (Brown et al., 2020) and LLaMA-63B (Touvron et al., 2023a;b) with billions of trainable parameters. Specifically, such a DP training system must have high time and memory efficiency, low communication cost, and the compatibility with general neural network architectures.

For small to moderately large models (e.g. with less than a billion parameters) that fit within the memory of a single GPU, a range of DP algorithms are feasible, producing the same result at different efficiency. Examples include TensorFlow-privacy Subramani et al. (2021), Opacus Yousefpour et al. (2021); Bu et al. (2022b), ghost clipping (GhostClip) Goodfellow (2015); Li et al. (2021); Bu et al. (2022a), and Book-Keeping (BK) Bu et al. (2023b), among which the BK algorithm has allowed DP optimization to be almost as efficient as the standard one. To be specific, the time/space complexity of BK algorithm is $1.08 \times /1.05\times$ of the standard optimization on ViT-Large (300M parameters, 147 layers) and $1.03 \times /1.01\times$ on GPT2-large (800M parameters, 220 layers).

To enable the DP distributed learning of these not-too-large models, one can directly use DDP (distributed data parallelism) (Li et al.), where each mini-batch of data is partitioned to smaller

micro-batches and each GPU computes one micro-batch with a full copy of the DP model. A line of researches (Yousefpour et al., 2021; De et al., 2022; Kurakin et al., 2022) have reported that DDP with DP usually either incurs huge memory cost due to caching the per-sample gradients, or suffers from $2 - 9\times$ slower training speed than non-DP optimization De et al. (2022); Bu et al. (2021). While the efficiency issues can be addressed through a better DP algorithm, such as BK, the feasibility issue remains insurmountable because DDP cannot train models that exceed the capacity of one GPU. Notably, the efficiency of BK algorithm is enhanced by two key techniques: *mixed ghost norm* (computing per-sample gradient norms almost for free) and *book-keeping trick* (only using one round of full back-propagation, not two rounds as in Li et al. (2021); Bu et al. (2022a)), which are detailed in Appendix A and will also be leveraged in our DP-ZeRO solution.

As the model size further increases beyond a reasonable bound for one GPU (e.g. 32GB memory, which roughly translates to 2B model training with Adam), the model must be partitioned in addition to the data, e.g. using pipeline parallelism and model parallelism, so that each GPU only holds a partial shard of the model (see Figure 2). In He et al. (2022), DP is combined with pipeline parallelism to fine-tune about 0.1% of GPT3-175B. Yet, the pipeline parallelism can be inefficient due to a non-DP-related issue, known as the pipeline bubble, where GPUs are idle while waiting for data to process.

Table 1: Summary of DP distributed learning.

| Distributed solution | Parallelism | Model sharding | Standard version | DP version | Remark |
|---|---|---|---|---|---|
| DDP | Data | No | Li et al. | Yousefpour et al. (2021) | unable to fit large model and DP is memory costly |
| DDP | Data | No | Frostig et al. (2018) | De et al. (2022) | unable to fit large model and DP is slow |
| GPipe | Pipeline | Yes | Huang et al. (2019) | He et al. (2022) | pipeline bubble wastes GPU time |
| ZeRO | Data(&Model) | Yes | Rajbhandari et al. (2020) | **Ours** | speed & memory & communication efficient |

Generally speaking, more advanced distributed methods such as Zero Redundancy Optimizer Rajbhandari et al. (2020) (**ZeRO**) and mixed-precision training have not be paired with DP due to the lack of algorithmic advances. In this work, we develop DP-ZeRO, equipping state-of-the-art distributed learning solution with DP (see comparison in Table 1), without altering the mathematics of DP optimization. We summarize our contributions as follows.

1. We propose the zero redundancy distributed learning with differential privacy (DP-ZeRO), demonstrating the same level of **communication efficiency**, **computation efficiency (speed and memory)**, and **scalability** (e.g. to GPT3 level and hundreds of GPUs) as the standard ZeRO.

2. We enable the mixed-precision training with DP by addressing the issue of loss scaling. This advance allows us to reduce the memory cost by roughly 50% and allow significantly faster communication that was previously not enjoyed by DP distributed learning.

3. We enable DP deep learning with more than 1B trainable parameters for the first time. E.g. we are the first to train the full GPT2-XL, ViT-Gigantic, ViT-10B and GPT-100B with DP.

4. We will open-source our codebase that automatically applies DP-ZeRO for general tasks (e.g. classification and language understanding), general network architectures (e.g. ResNet, ViT, GPT), and general distributed solutions (including DeepSpeed and FSDP), with one line of code change.

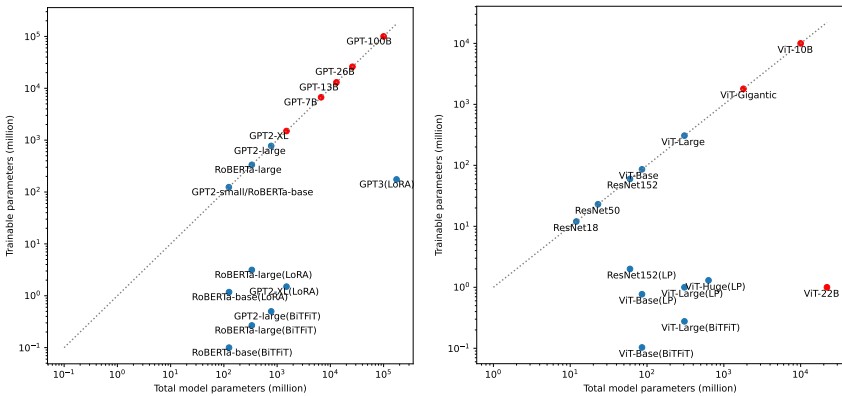

Figure 1: Total/trainable parameters of existing DP models (blue) and ours by DP-ZeRO (red).

## 2 PRELIMINARY

### 2.1 DIFFERENTIAL PRIVACY

DP provides a formal privacy guarantee, making it difficult to extract any information from training data. The privacy guarantee is characterized by $(\epsilon, \delta)$-DP in Definition 2.1, with smaller $(\epsilon, \delta)$ representing lower privacy risk.

**Definition 2.1** (Dwork et al. (2006)). A randomized algorithm $M$ is $(\varepsilon, \delta)$-DP if, for any two neighboring datasets $S, S'$ that differ by one sample and for any event $E$, we have $\mathbb{P}[M(S) \in E] \leqslant \mathrm{e}^\varepsilon \mathbb{P}[M(S') \in E] + \delta$.

In DP deep learning, the gradients are made private by post-processing through per-sample gradient clipping and random noising:

$$\text{private gradient: } \mathbf{G}_{[m]} := \sum_i C_i(R_m)\boldsymbol{g}_{[m],i} + \sigma_{\mathrm{DP}}\|[R_1, \cdots, R_M]\| \cdot \mathcal{N}(0, \mathbf{I}), \qquad (1)$$

Here the gradient of all trainable parameters is partitioned into $M$ groups, i.e. $\boldsymbol{g}_{[m],i}$ is the $i$-th per-sample gradient of the $m$-th group's parameters, where $m \in \{1 \cdots M\}$ is the group index. $C_i$ is the per-sample gradient clipping factor so that $\|C_i\boldsymbol{g}_{[m],i}\| \leq R_m$ and $R_m$ is the clipping threshold. That is, DP optimization is enabled when the standard optimizers such as stochastic gradient descent (SGD) and Adam (Kingma & Ba, 2014) update the trainable parameters with the private gradient, instead of the standard gradient $\sum_i \boldsymbol{g}_i$.

**Mathematical gradient partition** In (1), the trainable parameters and their gradients are mathematically partitioned into $M$ groups, e.g. in all-layer clipping, all parameters form one group ($M = 1$) Abadi et al. (2016); in layer-wise clipping McMahan et al. (2018); He et al. (2022), each layer's parameters form a group ($M$ = number of layers). Empirical evidence and theoretical analysis show that different partitions have the same training speed, though a finer partition (e.g. layer-wise) has lighter memory footprint[1].

**Per-sample gradient clipping** In (1), a number of clipping functions $C_i = C(\|\boldsymbol{g}_i\|; R)$ are available. Most works Abadi et al. (2016); Li et al. (2021); Yu et al. (2021) use the vanilla clipping $C_i = \min(R/\|\boldsymbol{g}_i\|, 1)$. Recently, Bu et al. (2023a); Yang et al. (2022) advocate the automatic clipping $C_i = 1/(\|\boldsymbol{g}_i\| + 0.01)$ which is hyperparameter-free and comparably accurate. Note if $C_i \equiv 1$, then the clipped gradient reduces to the standard gradient. The main overhead of DP optimization is the computation of per-sample gradient norms. On a single GPU, the mixed ghost clipping Bu et al. (2023b) has reduced the time complexity to $< 10\%$ on large models like GPT2.

**Privacy accounting** In (1), adding Gaussian noise to the clipped gradient protects the privacy that is quantifiable by the privacy accounting theory Abadi et al. (2016); Bu et al. (2020); Dong et al. (2019); Zhu et al. (2022); Gopi et al. (2021); Koskela et al. (2020). The privacy guarantee is increasing in the noise level $\sigma_{\mathrm{DP}}$, independent of $R_m$, learning rate, clipping function and model architectures, with $\sigma_{\mathrm{DP}} = 0$ leading to $\epsilon = \infty$ (non-private).

### 2.2 ZERO REDUNDANCY OPTIMIZER (ZERO)

#### 2.2.1 PARALLEL COMPUTING

Parallel computing is necessary to train large-scale models and is critical to the optimization efficiency. For models that fit in a single GPU, data parallelism (DataP) can be used to speed up the training by partition the mini-batch of samples into multiple micro-batches. Then, each GPU (holding a full copy of parameters) executes the forward and backward propagation of one micro-batch, from which the parameter gradients are generated and averaged across GPUs to update the trainable parameters. However, for models that do not fit in a single GPU, the model parameters need to be sharded by alternative solutions such as ZeRO Rajbhandari et al. (2020), model parallelism (ModelP) and pipeline parallelism (PipeP).

---

[1]We note that DP optimization under different group-wise clippings can have the same computation and communication efficiency (under the BK algorithm), with or without ZeRO.

ModelP partitions a model vertically, e.g. using 3 GPUs to store the parameters of one layer. As a consequence, ModelP does not scale efficiently beyond a single node due to fine-grained computation and expensive communication between layers. Implementation-wise, ModelP frameworks usually require heavy code integration that may not be generalizable in model architectures. In contrast, PipeP partitions a model horizontally across layers, e.g. storing 3 layers in each GPU. Each GPU deals with all micro-batches sequentially, though PipeP can be inefficient due to the pipeline bubble, which is overcome by ZeRO Rajbhandari et al. (2020). ZeRO is an advanced data parallel method that eliminates memory redundancies during the training, and improves the training speed and communication volume proportionally to the number of GPUs. Unlike basic DataP, ZeRO partitions a model's states across GPUs and gather/reduce in a just-in-time manner, thus sustaining the high efficiency of very large model training. Notice that ZeRO can work compatibly with ModelP and optionally offload the model states to CPUs Ren et al. (2021); Rajbhandari et al. (2021).

### 2.2.2 MODEL STATE PARTITION

A major part of the training memory is consumed by the model states[2]. ZeRO has three stages (ZeRO1/2/3) that partition these model states by different levels, with lower level being faster but more memory costly. For instance, in Table 2 of (Rajbhandari et al., 2020), ZeRO1/2/3 at most train 7.6/14.4/128B models on 64 V100 GPUs.

We take an example of mixed-precision Adam optimizer to train a model with $\Psi_{\text{model}}$ parameters, which maintains a master copy (fp32) of optimizer states , and the half-precision parameters and gradients.

**Optimizer state partition** The optimizer states are the (master) parameters, variance and momentum, each taking $4\Psi_{\text{model}}$ memory. ZeRO1 only applies the optimizer state partition and updates the parameters locally, reducing the memory cost of model states from $16\Psi_{\text{model}}$ for basic DataP to $(4 + \frac{12}{N_d})\Psi_{\text{model}}$ at each of $N_d$ GPUs.

**Hardware gradient partition** In addition to ZeRO1, during the back-propagation, ZeRO2 (and ZeRO3) further partitions the $2\Psi_{\text{model}}$ gradients into different GPUs, reducing the memory cost to $(2 + \frac{14}{N_d})\Psi_{\text{model}}$ at each GPU. Figure 2 illustrates the difference between the hardware partition and the mathematical partition in Section 2.1.

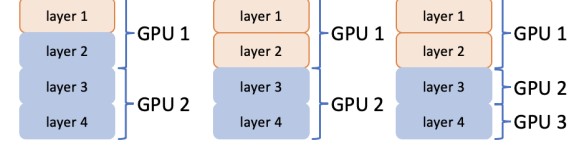

Figure 2: Mathematical (left two, same GPU allocation, different accuracy) and hardware (right two, different GPU allocation, same accuracy) gradient partition. Orange and blue are gradient groups $\boldsymbol{g}_{[1]}$ and $\boldsymbol{g}_{[2]}$ in DP optimization (1).

**Parameter partition** In addition to ZeRO2, ZeRO3 also partitions the $2\Psi_{\text{model}}$ fp16 parameters, further reducing the memory cost to $\frac{16}{N_d}\Psi_{\text{model}}$ at each GPU.

### 2.3 MIXED-PRECISION TRAINING

Mixed-precision training (Micikevicius et al., 2018) performs the forward and backward propagation on the half precision (fp16 or bf16) parameters, activations, and gradients, while performing the model update in full precision (fp32). Compared to the full-precision training, it is capable of saving the memory by $\approx 50\%$ and accelerating the computation by $\approx 20\%$. We note that fp16 has better precision but a limited range: its representable numbers are among $10^{-8} \sim 10^5$, and vice versa for bf16. Therefore, loss scaling is necessary when using fp16 to prevent small gradients from being rounded to zero, thereby preserving the model's accuracy (see Appendix C for details). In contrast, bf16 usually does not need loss scaling since it has the same range as fp32. However, bf16 is not as widely supported as fp16, e.g. only available on NVIDIA Ampere GPUs or above (Nvidia).

---

[2]Another important part of memory consumption is the batch-size-related variables such as the activation tensors, which is instantiated during the forward propagation and independent to DP (which modifies the gradient during the back-propagation).

## 3 DIFFERENTIALLY PRIVATE ZERO

### 3.1 ALGORITHM

Our DP-ZeRO algorithm introduces the per-sample gradient clipping and noising to the standard ZeRO (Rajbhandari et al., 2020), while maintaining the efficiency. At high level, an iteration of ZeRO consists of the following steps:

$$\left( \boxed{\text{all-gather}} \rightarrow \boxed{\text{forward}} \rightarrow \boxed{\text{all-gather}} \rightarrow \boxed{\text{backward}} \rightarrow \boxed{\text{reduce}} \right)^{\times L \text{ layers}} \rightarrow \boxed{\text{update(SGD/Adam/...)}}$$

The operations in $\boxed{\text{purple}}$ are global and require communication among GPUs, whereas the operations in $\boxed{\text{green}}$ are locally computed within each GPU. In particular, DP optimization is only different from the standard optimization in the back-propagation, which can be decomposed into

$$\boxed{\text{backward}} = (\text{output gradient} \rightarrow \text{clipping factor} \rightarrow \text{parameter gradient} \rightarrow \text{noising})$$

To give more details, we consider training a neural network of linear layers using $N_d$ GPUs. We emphasize that the following procedure is sufficiently generic to cover other layer types, such as convolution, embedding, normalization, and so on, which are all supported by DP-ZeRO. The full algorithm is depicted in Figure 3, where we denote the $j$-th micro-batched variables like $\boldsymbol{a}_l^{(j)}$, for $1 \leq j \leq N_d$.

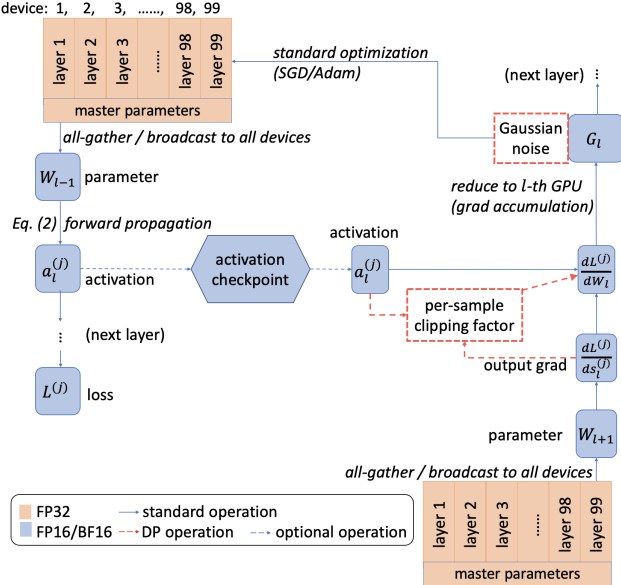

Figure 3: Algorithm of DP-ZeRO with mixed-precision training.

The forward propagation of DP optimization is the same as that of the standard optimization:

$$\boldsymbol{s}_l = \boldsymbol{a}_l \mathbf{W}_l + \mathbf{b}_l, \boldsymbol{a}_{l+1} = \phi_l(\boldsymbol{s}_l). \tag{2}$$

At the $l$-th layer, $\boldsymbol{a}_l \in \mathbb{R}^{BT_l d_l}$ is the layer's input, also known as the activation, $\boldsymbol{s}_l \in \mathbb{R}^{BT_l p_l}$ is the layer's output, $\mathbf{W}_l \in \mathbb{R}^{d_l p_l}$ is the weight, $\mathbf{b}_l \in \mathbb{R}^{p_l}$ is the bias, and $\phi_l$ is any inter-layer operation such as ReLU, tanh or pooling. We denote $B$ as the physical micro-batch size[3] and $T_l$ as the hidden feature dimension (e.g. sentence length or number of pixels). During the forward propagation, the activations $\{\boldsymbol{a}_l\}$ are computed and stored in the computation graph, and the loss $L = \sum_i L_i$ is derived, where $L_i$ is the per-sample losses. During the back-propagation, the output gradient is first computed based on the previous layer.

$$\frac{\partial L}{\partial \boldsymbol{s}_l} = \frac{\partial L}{\partial \boldsymbol{s}_{l+1}} \frac{\partial \boldsymbol{s}_{l+1}}{\partial \boldsymbol{a}_{l+1}} \circ \frac{\partial \boldsymbol{a}_{l+1}}{\partial \boldsymbol{s}_l} = \frac{\partial L}{\partial \boldsymbol{s}_{l+1}} \mathbf{W}_{l+1} \circ \phi_l'(\boldsymbol{s}_l),$$

---

[3]The micro-batch size $B$ is the number of samples processed by each GPU, which determines the time and memory efficiency, but not the performance. The logical batch size that determines the performance is $B \times N_d \times \text{gradient\_accumulation\_steps}$.

in which $\circ$ is element-wise multiplication. Specifically, the use of parameter $\mathbf{W}_{l+1}$ necessitates the all-gather operation when the model is partitioned into multiple GPUs, which is not needed in single GPU training. Next, the activation $\boldsymbol{a}_l$ is used together with $\frac{\partial L}{\partial \boldsymbol{s}_l}$ to compute the parameter gradient:

$$\text{DP gradient: } \frac{\partial \sum_i C_i L_i}{\partial \mathbf{W}_l} + \sigma \mathcal{N}(0, \mathbf{I}) = \boldsymbol{a}_l^\top \text{diag}(C_1, \cdots, C_B) \frac{\partial L}{\partial \boldsymbol{s}_l} + \sigma \mathcal{N}(0, \mathbf{I}).$$

Note the standard gradient can be viewed as $C_i = 1, \sigma = 0$. Here the per-sample gradient norm (or the clipping factor $C_i$) can be computed at small cost, as we have discussed in Section 1.

## 3.2 TIME EFFICIENCY OF DP-ZeRO

The time efficiency of DP-ZeRO consists of two parts: the local computation (including forward and backward propagation) and the global communication (including intra-node and inter-node communication). Given that the only difference between DP-ZeRO and ZeRO is the back-propagation, we claim that DP-ZeRO could enjoy high efficiency on-par with the standard ZeRO when (I) DP back-propagation exhibits a time efficiency comparable to the standard, similar to the single GPU training, and/or (II) the time efficiency of the parts other than back-propagation is not insignificant. We give the time of each part of DP-ZeRO in (3) to illustrate our claim.

$$\frac{\text{DP-ZeRO speed}}{\text{standard ZeRO speed}} = \frac{\textcolor{blue}{\text{back-propagation}} + \textcolor{green}{\text{forward propagation}} + \textcolor{orange}{\text{communication}}}{\textcolor{red}{\text{DP back-propagation}} + \textcolor{green}{\text{forward propagation}} + \textcolor{orange}{\text{communication}}} \quad (3)$$

To be explicit, we summarize the time complexity in Table 2 and refer to Appendix B for details.

Table 2: Time complexity of one iteration in distributed learning[4]. We denote $\Psi_{\text{train}}$ to be the number of trainable parameters ($\Psi_{\text{train}} = \Psi_{\text{model}}$ in full parameter training), and define $B, T$ below (2).

|  | forward propagation | | back-propagation | | | communication |
|---|---|---|---|---|---|---|
|  | activation | attention | output grad | param grad | DP clip and noise | |
| complexity | $2BT\Psi_{\text{model}}$ | $O(BT^2)$ | $2BT\Psi_{\text{model}}$ | $2BT\Psi_{\text{train}}$ | $0.666BT\Psi_{\text{train}}$ | $O(\Psi_{\text{model}})$ |

In what follows, we analyze the absolute and relative speed (to standard ZeRO) of DP-ZeRO under important settings.

### 3.2.1 NUMBER OF COMPUTATION DEVICES

When scaling from one GPU (zero communication) to one node (multiple GPUs) and to multiple nodes, the communication efficiency decreases sub-linearly ($\textcolor{orange}{\uparrow\text{communication}}$). On a single node, multiple GPUs can communicate using the high-speed intra-node connections such as NVLink/NVSwitch (Foley & Danskin, 2017; Ishii et al., 2018). On multiple nodes, which are necessary for large models, the inter-node connections are $3 \sim 24\times$ slower than the intra-node connections (Li et al., 2019; Zhang et al., 2022). In short, DP-ZeRO can be as fast as ZeRO by (3) when multiple nodes are employed.

### 3.2.2 MEMORY-EFFICIENT DISTRIBUTED LEARNING

The communication volume is specific to different distributed algorithms, most of which trade the communication or speed for memory, in order to feasibly train very large models. For example, ZeRO3 (but not ZeRO1/2) needs to all-gather the sharded parameters at each iteration, hence suffering from 50% extra communication volume ($\textcolor{orange}{\uparrow\text{communication}}$). Another example is the activation check-pointing (also known as gradient check-pointint Chen et al. (2016)), where a second forward propagation re-computes the expensive activations during back-propagation, though at a 33% slower speed ($\textcolor{green}{\uparrow\text{forward propagation}}$). These techniques improve the relative speed of DP-ZeRO but worsens the absolute speed.

---

[4]Here 0.666 is figurative and dependent on settings. Notice that $\Psi_{\text{train}} \ll \Psi_{\text{model}}$ when most parameters are frozen.

### 3.2.3 PARAMETER EFFICIENT FINE-TUNING

Parameter efficient fine-tuning (PEFT), such as LoRA (Hu et al., 2021), Adapter (Houlsby et al., 2019), and BiTFiT (Zaken et al., 2022), optimizes a small fraction (e.g. $\Psi_{\text{train}} = 0.1\%\Psi_{\text{model}}$) of model parameters and thus boosts the efficiency of back-propagation and communication ($\downarrow$DP back-propagation $\downarrow$back-propagation $\downarrow$communication). Consequently, (I) the communication volume of the gradient can be reduced possibly by $1000\times$; (II) the local computation can accelerate by 50% (Hu et al., 2021; Bu et al., 2022b), which can be seen by treating $\Psi_{\text{train}}$ in Table 2 as almost zero; (III) the memory cost is saved on the non-trainable layers, which translates to larger batch size and faster computation. Hence, both relative and absolute speed of DP-ZeRO improve using PEFT.

## 3.3 MEMORY EFFICIENCY OF DP-ZERO

We claim that DP-ZeRO is as memory efficient as the standard ZeRO, similar to the single GPU training, when we use (I) the mixed ghost norm trick Bu et al. (2022a; 2023b), instead of GhostClip (Goodfellow, 2015; Li et al., 2021) or per-sample gradient instantiation (Yousefpour et al., 2021); (II) the layer-wise clipping style instead of the all-layer clipping, so that the book-keeping (Bu et al., 2023b) does not store all output gradients; (III) a large number of GPUs so that the micro-batch size $B$ (i.e. per-GPU batch size) is small: specially, when $B = 1$ in the gradient accumulation, the per-sample gradient is free. We empirically verify our claim in Figures 5 and 7.

## 3.4 MIXED-PRECISION TRAINING WITH DP

We now analyze the intricacy in mixed-precision training with DP, which is not unique to DP-ZeRO but present in the general DP optimization. We emphasize that the per-sample gradient clipping already plays the role of scaling, and hence DP mixed-precision training must not use loss scaling, as illustrated in Table 3. Specifically, in standard mixed-precision training, there are two steps of scaling: (I) scaling up the loss $L_i$ by $10^3 \sim 10^9$ (and consequently the output gradient $\frac{\partial L_i}{\partial s}$ as well as $\frac{\partial L_i}{\partial \mathbf{W}}$) before the back-propagation, to prevent the underflow where fp16 gradient is too small to be distinguished from 0, and (II) scaling down the parameter gradient $\frac{\partial L_i}{\partial \mathbf{W}}$, by the same factor, after the back-propagation to recover the correct magnitude of gradient.

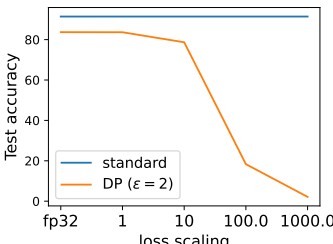

Figure 4: Accuracy of mixed-precision training with loss scaling. ViT-large on CIFAR100.

However, in DP mixed-precision training, scaling up the loss may cause overflow, while scaling down the gradient incorrectly over-shrinks the gradient and worsens the performance. See Figure 4 for a real example. We explain this intricacy step-by-step in Appendix C.

Table 3: Illustration of overflow and underflow issues during mixed-precision training (ghost norm).

| loss scale=$10^3$ | activation $\boldsymbol{a}_l$ | output grad $\frac{\partial L}{\partial s_l}$ (scaled) | per-sample grad norm | | clipping factor | param grad (not scaled down) | param grad (if scaled down) |
|---|---|---|---|---|---|---|---|
| | | | vec($\boldsymbol{a}_l\boldsymbol{a}_l^\top$) | vec($\frac{\partial L}{\partial s_l}\frac{\partial L}{\partial s_l}^\top$) | | | |
| standard w/o scaling | $10^{-3} \sim 10^2$ | $10^{-8} \sim 10^1$ | N/A | N/A | 1 | $10^{-7} \sim 10^1$ | $10^{-7} \sim 10^1$ |
| standard w/ scaling | $10^{-3} \sim 10^2$ | $10^{-5} \sim 10^4$ | N/A | N/A | 1 | $10^{-4} \sim 10^4$ | $10^{-7} \sim 10^1$ |
| DP w/o scaling | $10^{-3} \sim 10^2$ | $10^{-8} \sim 10^1$ | $10^2 \sim 10^3$ | $10^{-6} \sim 10^0$ | $10^{-3} \sim 10^2$ | $10^{-7} \sim 10^1$ | $10^{-7} \sim 10^1$ |
| DP w/ scaling | $10^{-3} \sim 10^2$ | $10^{-5} \sim 10^4$ | $10^2 \sim 10^3$ | $10^0 \sim 10^6$ | $10^{-6} \sim 10^{-1}$ | $10^{-7} \sim 10^1$ | $10^{-10} \sim 10^{-2}$ |

## 4 EMPIRICAL PERFORMANCE OF DP-ZERO

We evaluate DP-ZeRO on five aspects: model architectures, efficiency, scalability, compatibility with various distributed learning and DP techniques. We use DP-ZeRO to train ResNet (He et al., 2016), ViT (Dosovitskiy et al., 2020; Zhai et al., 2022) and GPT (Radford et al., 2019; Brown et al., 2020), which are workhorses in computer vision and language tasks. We measure the time and memory efficiency of DP-ZeRO under settings such as PEFT and multiple precision formats (fp32 or fp16/bf16). We evaluate the scalability of DP-ZeRO in terms of number of GPUs and number of model parameters. Our experiments scale from single node (8 GPUs) to multiple nodes, up to 256 GPUs, and train models up to 100B trainable parameters. Moreover, DP-ZeRO is compatible

with mainstream implementations of ZeRO[5] and with different clipping styles, clipping functions, privacy accountants, and so on. We leave the experimental details in Appendix D. By default, we use AdamW, mixed-precision training, layer-wise clipping style, $B = 4$, and A100 GPU with 40GB memory, unless otherwise stated.

## 4.1 GENERALITY OF DP-ZERO

DP-ZeRO is generally applicable to different neural network architectures, clipping styles, and precision formats. We test DP-ZeRO1 on single node and observe that different clipping styles are equally fast, but layer-wise clipping is more memory efficient than all-layer clipping. Comparing to the standard ZeRO, our DP-ZeRO enjoys almost the same speed and memory efficiency, while the gap will be further closed as we move to more advanced stages of ZeRO.

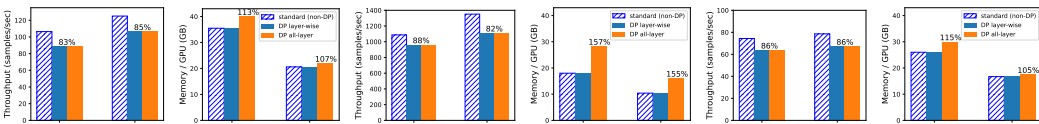

Figure 5: Efficiency on ViT-Gigantic (left, 1.8B), ResNet152 (middle) and GPT2-XL (right, 1.5B).

## 4.2 LIGHTER TRAINING OF DP-ZERO

DP-ZeRO can employ low-memory optimizers and train on fewer parameters, therefore vastly reducing the memory and communication cost. On a single node, we demonstrate that DP-ZeRO actually benefits (more than standard ZeRO and single-GPU training) from lighter training, following from our discussion in Section 3.2.

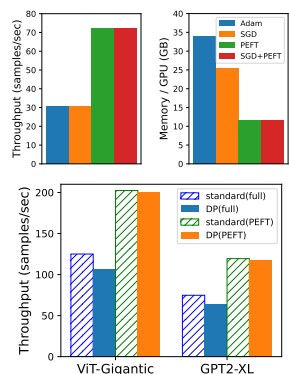

**Low-memory optimizers** Low-memory optimizers can boost the training efficiency at the cost of accuracy degradation. For example, SGD requires only 1/3 optimizer states of Adam and significantly saves the memory; 1-bit Adam (Tang et al., 2021) and signSGD compress the gradient and reduce the communication volume up to $32\times$.

**Fewer trainable parameters** In the fine-tuning phase, as we analyzed in Section 3.2.3, PEFT improves both the local computation and the communication volume. Hence, DP-ZeRO allows PEFT on ViT and GPT to be $\approx 2\times$ faster than full fine-tuning, whereas the single GPU acceleration is $\leq 1.5\times$.

Figure 6: Efficiency of DP-ZeRO with lighter training. Upper: $> 2\times$ speedup with lighter training; ViT-5B, $B = 1$. Lower: DP-ZeRO benefits more than ZeRO from PEFT.

**Remark 4.1.** We leverage DP-ZeRO3 with SGD to train ViT-10B (full parameters) and ViT-22B (PEFT; 1M trainable parameters) on one node. See Appendix D for configurations.

## 4.3 THREE STAGES OF DP-ZERO

DP-ZeRO supports all stages of ZeRO under different implementations including DeepSpeed (default) and FSDP.

In Figure 7, the efficiency of DP-ZeRO catches up with the standard ZeRO when we move up the stages. For instance of ViT-Gigantic, the throughput increases from 83% by DP-ZeRO1 to 95-97% by DP-ZeRO3. Following (3), we can attribute the relatively fast training of DP-ZeRO to the increase cost of communication, especially in DP-ZeRO3. Additionally, we observe that the throughput of DP-ZeRO1/2 improves to over 95% on 4 nodes, as predicted by Section 3.2.1. Notice that we save DP-ZeRO3 of GPT to Section 4.4 on super-large scale.

---

[5]DP-ZeRO is implemented on DeepSpeed (supporting ZeRO1/2/3), FSDP (Zhao et al.) (supporting ZeRO3), MiCS (Zhang et al., 2022) (supporting ZeRO2/3), and any distributed optimizers supported on them.

Figure 7: Efficiency of DP-ZeRO on ViT-Gigantic and GPT2-XL under different implementations.

## 4.4 SCALABILITY OF DP-ZERO

We evaluate the scalability of DP-ZeRO3 in terms of large sequence length (2048), large model size ($7 \sim 100B$), and large number of GPUs (up to 256). We use A100 with 80GB memory, as well as the activation check-pointing and ModelP.

In Figure 8 (left), we observe that for a fixed model with 26B trainable parameters, DP-ZeRO is super-linearly scalable to the number of GPUs, achieving $> 95\%$ speed of the standard ZeRO. Here super-linearity is a property of ZeRO (see Figure 3 in Rajbhandari et al. (2020)) which allows more GPUs to shard the model states (and reduce the per-GPU memory cost) more aggressively, and to train faster since the micro-batch size is larger. Furthermore, in Figure 8 (right), for a fixed number of GPUs, DP-ZeRO is linearly scalable to the model size, achieving the same speed as the standard ZeRO. In short, DP-ZeRO is almost equal to ZeRO in terms of training efficiency in super-large scale.

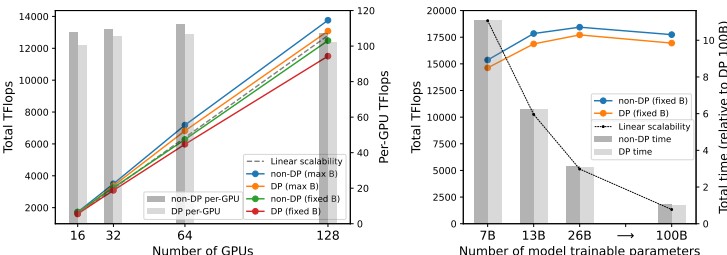

Figure 8: Scalability of DP-ZeRO3 on 26B model (left) and 256 GPUs (right). 'max B' means we fit the maximum micro-batch in each GPU and 'fixed B' means $B = 2$.

**Remark 4.2.** In comparison to DP-ZeRO, DataP (DP or standard) at most fits 5B models (Rajbhandari et al., 2020) in 80GB memory, regardless of the number of GPUs. We cannot compare to DP-PipeP in He et al. (2022) because the codebase and experiment details (e.g. number of trainable parameters and sequence length) are not publicly available. Nevertheless, since DP-ZeRO resembles the efficiency of standard ZeRO, it suffices to demonstrate the usefulness of DP-ZeRO by comparing ZeRO to PipeP.

## 5 DISCUSSION

In this work, we develope DP-ZeRO that enables the optimization of models up to 100B trainable parameters, thus allowing DP distributed learning to be as efficient and scalable as the standard one. We believe this is a significant milestone to pave the path towards DP foundation models, especially for its open-source nature (link to be released).

We emphasize that, since DP only modifies the back-propagation, our DP-ZeRO is orthogonal to any large-scale training techniques that are not tied to back-propagation: for example, activation check-pointing, CPU offloading, weight/activation quantization (Dettmers et al., 2023; Xiao et al., 2023), tensor parallelism (Narayanan et al., 2021) and other techniques yet to come. Note that to make fair comparisons in this paper, DP and non-DP optimization are implemented without fusing the operations such as tensor multiplications. Therefore, the efficiency of DP deep learning can be further and ultimately improved, from an engineering perspective, by implementing the DP back-propagation with operator fusion via CUDA kernels and C++ coding.

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

# A   THE BOOK-KEEPING (BK) ALGORITHM

## A.1   EFFICIENT COMPUTATION OF PER-SAMPLE GRADIENT NORMS

The mixed ghost norm Bu et al. (2022a) is the state-of-the-art technique to compute the per-sample gradient norm of the **weight**, almost for free. It hybridizes two basic techniques – the per-sample gradient instantiation and the ghost norm – to compute the Frobenius norm of weight gradient,

$$\left\| \boldsymbol{a}_{l,i}^\top \frac{\partial L}{\partial \boldsymbol{s}_{l,i}} \right\|^2 \overset{\text{per-sample grad}}{=\!=\!=\!=\!=} \left\| \frac{\partial L_i}{\partial \mathbf{W}_l} \right\|_{\text{Fro}}^2 \overset{\text{ghost norm}}{=\!=\!=\!=\!=} \operatorname{vec}\left( \frac{\partial L}{\partial \boldsymbol{s}_{l,i}} \frac{\partial L}{\partial \boldsymbol{s}_{l,i}}^\top \right) \cdot \operatorname{vec}(\boldsymbol{a}_{l,i} \boldsymbol{a}_{l,i}^\top) \qquad (4)$$

where "vec" flattens the tensor to an one-dimensional vector. In words, Equation (4) gives two equations that are equivalent mathematically, but significantly different in efficiency:

- $\|X\|_{\text{Fro}}^2 = \left\| A^\top B \right\|^2$ firstly computes $A^\top B$ and then its norm.
- $\|X\|_{\text{Fro}}^2 = \operatorname{vec}\left( AA^\top \right) \cdot \operatorname{vec}(BB^\top)$ firstly computes $AA^\top, BB^\top \in \mathbb{R}^{TT}$ and then their dot product.

In summary, the mixed ghost norm always applies the cheaper of two techniques at each layer of a neural network.

Finally, we note that the per-sample gradient norm of the **bias** is computed differently. This is because

$$\frac{\partial L_i}{\partial b_l} = \mathbf{1}^\top \frac{\partial L}{\partial \boldsymbol{s}_{l,i}}$$

is not actually a product of tensors like $X = A^\top B$. In fact, the multiplication with $\mathbf{1}$ turns out to be a summation along the first dimension, and it suffices to use per-sample gradient instantiation for the bias.

## A.2   BOOK-KEEPING THE OUTPUT GRADIENT

The BK algorithm uses two rounds of back-propagation (though each round only takes half the complexity, hence the total complexity of DP back-propagation matches the non-DP back-propagation). Therefore, output gradients $\frac{\partial L}{\partial \boldsymbol{s}_{l,i}}$ are kept to avoid repeated computation. Notice that the output gradient are relatively cheap to book-keep (see Figure 4 and Figure 5 in Bu et al. (2023b)).

# B   COMPONENT-WISE TIME COMPLEXITY OF DP-ZERO

In this section, we explain the time complexity of each part in Table 2, and demonstrate how the complexity can be different under different settings.

**Forward propagation:** The matrix multiplication during forward propagation results in $2BT\Psi_{\text{model}}$ complexity (see Bu et al. (2022a)). Notice that, if the activation check-pointing is used, essentially two rounds of forward propagation take place in one iteration. Hence the time complexity doubles and becomes $4BT\Psi_{\text{model}}$.

**Back-propagation:** This contains two sub-processes: the output gradients are computed at all layers, taking $2BT\Psi_{\text{model}}$ complexity; the parameter gradients are computed only at trainable layers (a few if doing PEFT), taking $2BT\Psi_{\text{train}}$ complexity. Clearly, in full parameter training, the total is $4BT\Psi_{\text{model}}$, and in PEFT, about $2BT\Psi_{\text{model}}$.

**Attention:** The time complexity of attention is $O(BT^2)$ in Vaswani et al. (2017), where $T$ is the sequence length (a.k.a. token length). When $T$ is large, e.g. training with long context like $T = 8192$, this cost is prohibitively high. In this regard, a line of researches have proposed linear complexity attention, including but not limited to Wang et al. (2020); Katharopoulos et al. (2020); Shen et al. (2021).

**Communication:** For algorithms that don't shard the model, such as data parallelism and ZeRO1/2, the communication is only used to send gradients and optimizer states. Hence the communication

volume is proportional to the number of trainable parameters $O(\Psi_{\text{train}})$. Otherwise, for algorithms such as ZeRO3 and tensor parallelism, the communication volume is proportional to the number of total parameters $O(\Psi_{\text{model}})$, because the forward propagation needs to gather the parameters from many GPUs. This makes a big difference in PEFT when $\Psi_{\text{train}} \ll \Psi_{\text{model}}$.

## C  LOSS SCALING IN MIXED-PRECISION TRAINING

We write the per-sample gradient with loss scaling $S$ as

$$\frac{\partial C_i L_i}{\partial \mathbf{W}_l} = C_i \frac{1}{S} \cdot \left( \boldsymbol{a}_{l,i}^\top \big( S \cdot \frac{\partial L}{\partial \boldsymbol{s}_{l,i}} \big) \right)$$

This covers the standard gradient ($C_i = 1$) and DP gradient (e.g. $C_i = 1/\|\boldsymbol{g}_i\|$, computed by the mixed ghost norm in Appendix A), in which $S$ enlarges the output gradient to avoid underflow, and $\frac{1}{S}$ shrinks the parameter gradient to the correct magnitude.

Recall that a standard mixed-precision training (with loss scaling) uses steps $1 \to 2 \to 3 \to 5 \to 6$ [6], or $1 \to 3 \to 6$ without loss scaling.

1. Forward propagation (fp16 weights and activations) and get the loss.

2. ~~Scaling up the loss by a factor $S$.~~

3. Backward propagation on the scaled loss (fp16 parameters and their gradients).

4. Per-sample gradient clipping (sensitivity $= 1$) and noising for DP.

5. ~~Scaling down the parameter gradient by a factor of $1/S$.~~

6. Update the parameters with their gradients.

If we follow the same procedure under the DP regime, say using a hook function to be called after back-propagation creates the gradients like in Opacus (Yousefpour et al., 2021), Private-Transformers (Li et al., 2021), FastDP (Bu et al., 2023b), then the per-sample clipping factor is scaled up $S$ times so as to normalize the gradient. Hence per-sample gradient clipping has already played the role of scaling down. If we scale down the gradient for a second time, the gradient is incorrectly over-shrunk. This is the case in Yu et al. (2021) and in the alternative implementation of (Li et al., 2021, Appendix T) (see also Figure 4). To be sure, this approach is still DP, but the performance does not match fp32 DP training correctly, and usually degrades too much to be useful.

One walk-around is to prevent per-sample gradient clipping to scale down the gradients and let step 5 do its job, i.e. $1 \to 2 \to 3 \to 4^* \to 5 \to 6$. We note that (Li et al., 2021, Appendix T) follows this path (though no experiment results or codes are available at the time of writing) by modifying step 4: clipping threshold (sensitivity)$= S$ instead of 1, so that the clipped gradient is $S$ times larger than the DP f32 training, to be scaled down by step 5. However, this introduces additional design decisions and does not prevent overflow when using fp16 (due to step 2, see Table 3).

Another walk-around is to delete step 5 and let per-sample gradient clipping do its job, i.e. $1 \to 2 \to 3 \to 4 \to 6$. However, this approach is harder to implement because in the standard process step 2 and 5 are simultaneously enabled or disabled. Also we cannot prevent overflow when using fp16 as we still use step 2.

Therefore, we propose to not use loss scaling (or equivalently we set $S = 1$ statically for all steps) during DP mixed-precision training, i.e. $1 \to 3 \to 4 \to 6$. Although, by not using step 2, we cannot prevent underflow when using fp16, this is much less a problem compared to overflow: underflow (treating small values as 0) makes the training less accurate but does not fail the training like overflow (treating large values as NAN). Lastly, the underflow issue is perfectly mitigated by bf16, which we recommend for DP mixed-precision training whenever possible .

---

[6]See                https://docs.nvidia.com/deeplearning/performance/
mixed-precision-training/index.html#lossscaling.

| | steps | fp16 issue | note | reference |
|---|---|---|---|---|
| standard | 136 | underflow | | Micikevicius et al. (2018) |
| standard | 12356 | none | | Micikevicius et al. (2018) |
| DP | 123456 | overflow | incorrect due to over-shrinking | Li et al. (2021) |
| DP | 1234*56 | overflow | different clipping threshold | Li et al. (2021) |
| DP | 1346 | underflow | perfect with bf16 | ours |
| DP | 12346 | overflow | hard to implement | ours |

Table 4: Mixed-precision training with DP or not.

## D  EXPERIMENT SETTINGS

Datasets: To evaluate the efficiency, it suffices to declare the data's dimension (e.g. micro-batch size and feature dimension) without specifying the dataset (though sometimes specifying the dataset means declaring the dimension, e.g. MNIST usually means 28*28 pixels). This is the norm in system papers such as Rajbhandari et al. (2020; 2021); Zhao et al.. In this work, vision models are trained with 224*224 pixels at ImageNet scale; GPT models are trained with sequence length 100, except in Figure 8 where sequence length is 2048.

Figure 4 and Table 3: We train ViT-large (300M parameters) and CIFAR100, 5 epochs, learning rate 5e-4, logical batch size 1000.

Figure 6: To fit as large a model as possible, we set $B = 1$ and use SGD. We set 48 attention heads, 21 layers, MLP=4*width (also known as embedding dimension), and modify width for all models. For instance, ViT-10B uses width=$768 * 22$, ViT-22B uses width=$768 * 34$.

Figure 8: We train AdamW with layer-wise clipping. DP distributed learning is based on MiCS (ZeRO3) using bf16 mixed-precision training. Most of GPT configuration is the same as Touvron et al. (2023a) (Table 2) in terms of embedding dimension, attention heads and number of layers. However, GPT-100B uses the configuration from Brown et al. (2020) (Table 2.1) but a smaller width.

## E  CODEBASE DESIGN

### E.1  WITH FORWARD & BACKWARD HOOKS

Hooks[7] are important functions to enrich the deep learning optimization. To be specific, there are

1. forward modular hook (nn.register_forward_hook),
2. backward modular hook (nn.register_backward_hook),
3. backward tensor hook (tensor.register_hook).

DP libraries including Opacus Yousefpour et al. (2021), Private-transformers Li et al. (2021), Private-Vision Bu et al. (2022a), FastDP Bu et al. (2023b;a), FastGradClip Lee & Kifer (2021) and so on, use modular hooks to modify the standard optimization. However, ZeRO libraries including DeepSpeed and FSDP use tensor hooks. This difference in the types of hooks and many other differences (e.g. both ZeRO libraries and DP libraries modify the optimizer's step function) cause non-trivial problems when combining DP with ZeRO. For example, to keep DP optimization as efficient as the standard, it is necessary to not waste time on computing the non-private gradient. However, if we skip such computation, then ZeRO's tensor hook will not be triggered and the corresponding distributed-learning-related operations cannot carry on. For another example, because DP and ZeRO add different types of hooks, the number of hooks is larger than either optimization and they slows down the training: consider an 100-layer network, each layer with weight and bias (2 tensors), then DP-ZeRO in this subsection needs 100 modular hooks and 200 tensor hooks, adding to a total of 300 hooks. In addition, the Book-Keeping algorithm (in FastDP) in its original form cannot be implemented together with ZeRO3, because all model states are partitioned including the output

---

[7]See https://pytorch.org/tutorials/beginner/former_torchies/nnft_tutorial.html#forward-and-backward-function-hooks.

gradients which are meant to be book-kept. To work around this requires rewriting the distributed solution's communication mechanism, and if successful, still requiring additional communication cost during the second back-propagation. Similar problems are present for Opacus and FastGradClip, which instantiates per-sample gradients that will be partitioned in ZeRO2/3 and requires additional communication cost when gathered to create the privatized gradient.

As a consequence, the hooks are fully supported on DP-ZeRO1 and partially supported on DP-ZeRO2/3 under the layer-wise clipping.

### E.2 WITHOUT HOOKS

Instead of registering hooks on top of the original (non-DP) back-propagation, we can directly modify the back-propagation following Appendix A: e.g., given the activation and output gradient,

$$
\frac{\partial C_i L_i}{\partial \mathbf{W}_l} = \boldsymbol{a}_{(l),i}^\top \frac{\partial L}{\partial \boldsymbol{s}_{(l),i}} \bigg/ \sqrt{\mathrm{vec}\left(\frac{\partial L}{\partial \boldsymbol{s}_{(l),i}} \frac{\partial L}{\partial \boldsymbol{s}_{(l),i}}^\top\right) \cdot \mathrm{vec}(\boldsymbol{a}_{(l),i} \boldsymbol{a}_{(l),i}^\top)}
$$

This approach requires rewriting the back-propagation for each layer type (linear, embedding, convolution, normalization, ...) and can be done at different levels (Pytorch, C++, CUDA kernel).

### E.3 USER INTERFACE

DP-ZeRO can be enabled by one piece of code: after the model is instantiated,

```
privacy_engine = PrivacyEngine(model,
                batch_size=256, sample_size=50000,
                epochs=3, target_epsilon=3)
```

The codebase is designed not to modify the optimizer, hence DP-ZeRO can work with arbitrary optimizer. Because of this design, our DP-ZeRO will not distinguish micro-batches. This is different from the gradient accumulation in Opacus (version ==0.x) and Private-Vision, where only the last micro-batch is processed by "optimizer.step()" but all other micro-batches are processed by "optimizer.virtual_step()". In other words, the noise $\sigma_{\mathrm{DP}} N(0, I)$ is added on the last micro-batch, after the micro-batches are accumulated. But DP-ZeRO adds the noise on each micro-batch equally. Note that the noise level per micro-batch is $\sigma_{\mathrm{DP}}/\sqrt{N_d}$ if a random seed is set across $N_d$ GPUs, or $\sigma_{\mathrm{DP}}/N_d$ otherwise.

