# OpenReview forum: "Zero redundancy distributed learning with differential privacy"
_ICLR.cc/2024/Conference — ICLR 2024 Conference Desk Rejected Submission_

### Official Review · Reviewer_zq48 · 2023-11-01

**Soundness:** 3 good
**Presentation:** 3 good
**Contribution:** 3 good
**Rating:** 8
**Confidence:** 2

**Summary:**

This paper proposes a novel distributed learning framework with differential privacy. The proposed method is demonstrated effectively on large models up to 10^5 million parameters. The algorithm is built upon ZeRO framework and several efficient DP training tricks from prior works. The experiments are thorough and the results are impressive.

**Strengths:**

1. This paper is well-written despite some format issues.
2. The proposed method enables training GPT-100B which beats state-of-the-art DP training in terms of scalability and efficiency.

**Weaknesses:**

I am curious about the utility of large models, like the GPT-100B. Is there a plot that shows accuracy drops?

**Questions:**

Please see weaknesses

---

> ### Author Response · Authors · 2023-11-13
> **Thank you!**
>
> We thank the reviewer for liking our paper. Since this paper is system-design focused, like the non-DP ZeRO paper (https://arxiv.org/abs/1910.02054), we didn't evaluate the accuracy. However, there is a series of work that demonstrates the better accuracy from larger models. E.g. GPT2-small (100M) to GPT2-large (800M) increases BLEU score from 61 to 64 in our first paragraph in Section 1. We are training and evaluating GPT-100B and believe this will be very interesting future work.

---

### Official Review · Reviewer_otjb · 2023-11-06

**Soundness:** 2 fair
**Presentation:** 2 fair
**Contribution:** 2 fair
**Rating:** 5
**Confidence:** 2

**Summary:**

The paper proposes DP-ZERO, a distributed training algorithm based on ZERO and adapted to do differentially private training with clipping and noising. The authors analyze the time and memory efficiency of DP-ZERO, showing that they're the same as original ZERO. The authors also conducted empirical evaluations on large models.

**Strengths:**

The paper tackles a very important problem as differentially private training of large models is becoming more and more needed in this era.
The empirical evaluations prove that the proposed algorithm can be used to train large models.

**Weaknesses:**

The presentation of the paper might need to be improved in order for readers to have a clearer understanding of the proposed algorithm and its significance.

1. Is DP-ZERO basically doing clipping in the unit of device, i.e. gradients of layers in one device is clipped? If so (or not), you might consider stating that more clearly.

2. One thing I found a bit missing is the difficulty of the problem, i.e. why is it non-trivial to make a DP version of the original ZERO? I kind of feel like it's quite straightforward to add clipping to the algorithm, especially if (1) is true as we can simply operate on each device individually for clipping, which is the same as non-distributed training.

3. The explanation of ZERO (Section 2) is a bit unclear at least to readers without much background knowledge like me. For example, in Section 2.2.2, you might state the memory needed for each variable clearly, so that readers can more easily add them up (e.g. to get the 16, 12, 14). You might state the unit of the memory (I presume "byte"?). Also, in Section 2.3, I think you can explain more details on mixed precision training, as that seems to be different for non-private and private training (as is mentioned in Section 3.4).

Minor: 2nd last sentence in Section 2.3: "does not need loss" -> "does not need loss scaling"?

4. The complexity of "DP clip and noise" in Table 2 seems to be a very important part of the paper and worth some detailed explanation in the main body. (I don't quite understand how it's calculated even after looking at Appendix B.) What does it mean to be "figurative and dependent on settings"?

**Questions:**

Besides the points mentioned above in "Weaknesses", I wonder if the precision change can affect the precision of the noise and thus causing any problem / difference in privacy.

---

> ### Author Response · Authors · 2023-11-13
> **Response Part1**
>
> We thank the reviewer for the comments and insightful questions. We will add our response to the camera-ready revision. Here is a point-to-point response.
>
> 1. Is DP-ZERO basically doing clipping in the unit of device, i.e. gradients of layers in one device is clipped? If so (or not), you might consider stating that more clearly.
>
> **Response** No. DP-ZeRO is capable of doing clipping (a mathematical operation) at various levels, e.g. all-layer or layer-wise, not limited to device-level. We refer the reader to paragraph "Mathematical gradient partition" in Section 2.1 and paragraph "Hardware gradient partition" in Section 2.2.2 for a detailed explanation. We also gave examples in Figure 5 to compare different clipping styles. That is, regardless of how the devices partition the layers, DP-ZeRO can perform flexible per-sample gradient clipping.
>
> To be concrete, say a model has 4 layers {1,2,3,4} and stores {1,2} in GPU1 and {3,4} in GPU2. Then device-level clipping computes and clips twice, on the gradient norms of layers {1,2} and those of layers {3,4}, respectively. Our DP-ZeRO can clip the all-layer gradients across devices, via the communication mechanism of ZeRO distributed solution.
>
> 2. Why is it non-trivial to make a DP version of the original ZERO?
>
> **Response** We emphasize it is highly non-trivial to make ZeRO working with DP. To give some context here, we highlight that large model training (which necessitates distributed learning) is different from its smaller counter-parts.
>
> a. Communication is necessary but complicated. Large models are partitioned and re-gathered during forward pass and back-propagation in the distributed training, which is not involved in non-distributed training. Because of the partition, it is delicate to impose DP within the complicated orchestra of communication. As a result, in Figure 3, we add the per-sample clipping *before* the communication and the noise addition *after* it, in order to maximize the speed and memory efficiency. There are many difficult cases in DP-ZeRO. For example, ZeRO3 hardly works with all-layer clipping because each layer is processed then thrown away in ZeRO (if we don't throw away, then the memory will explode; if we do throw away, then the clipped gradient requires a second back-propagation in ghost clipping by https://arxiv.org/abs/2110.05679, hence slowing down DP-ZeRO by 50\%);  extra care is needed for the calculation of noise level (see Appendix E.3).
>
> b. Mixed precision is tricky. Large models requires lots of memory that small models do not. Hence mixed precision training is highly desired to save about 50\% memory (see Figure 7), which is an important feature in DP-ZeRO. However, mixed precision could not work compatibly with DP at the time of writing. Naive combining DP with mixed precision results in wrong gradients and bad accuracy (see Section 3.4 and more explanation in Appendix C). We showed rigorously that loss scaling must not be used together with per-sample clipping. Only through a careful analysis can we achieve the both *numerical stability and accuracy* of DP mixed precision training.
>
> c. Technical difficulty. Coding up and open-sourcing DP-ZeRO is another main contribution of this work. We devote Appendix E to discuss the engineering challenges we have faced. For example, all existing Pytorch codebases (1) uses module hooks to impose DP and (2) modifies the optimizers. However, ZeRO --- a complicated library that already imposes its communication through these approaches will conflict with the DP operations. A significant amount of efforts has been put into making DP-ZeRO as efficient as the standard ZeRO.

---

> ### Author Response · Authors · 2023-11-13
> **Response Part2**
>
> 3. The explanation of ZERO (Section 2) is a bit unclear at least to readers without much background knowledge like me....Also, in Section 2.3, I think you can explain more details on mixed precision training,...
>
> **Response** We are happy to explain more about ZeRO, although we didn't do so in the first place due to page limit and the fact that this is from a published work (https://arxiv.org/abs/1910.02054). Essentially, there are five variables taking up the memory, (1)  fp32 parameters (2) fp32 variance (3) fp32 momentum (4) fp16/bf16 parameters (5) fp16/bf16 gradients. Each fp32 number takes 32 bits=4bytes memory, and each fp16/bf16 number takes 16bits=2bytes memory. Hence (1)=(2)=(3)=4$\Psi_{model}$ bytes and (4)=(5)=2$\Psi_{model}$. Next, different distributed solutions (including but not limited to ZeRO) partition the variables differently, if a variable is partitioned, then its memory occupancy is divided by the number of devices $N_d$. For instance, ZeRO1 only partition (1)(2)(3), so per-device memory is (4)+(5)+[(1)+(2)+(3)]/$N_d$=4+12/$N_d$.
>
> Note this only holds for Adam under mixed precision training. For instance, if we use SGD, then there is no (2) variance and maybe no (3) momentum; if we use PEFT, e.g. only optimizing 1\% parameters such as LoRA or only optimizing the bias terms, then (5)=1\% of (4) and (2)=(3)=1\% of (1). We discussed these variants in Section 3.2 and Section 4.2.
>
> Minor: 2nd last sentence in Section 2.3: "does not need loss" -> "does not need loss scaling"?
>
> **Response** Yes this is a typo. We will fix it.
>
> 4. The complexity of "DP clip and noise" in Table 2 seems to be a very important part of the paper and worth some detailed explanation in the main body. (I don't quite understand how it's calculated even after looking at Appendix B.) What does it mean to be "figurative and dependent on settings"?
>
> **Response** Table 2 is important but most results are known (from "Differentially Private Optimization on Large Model at Small Cost" Appendix B), e.g. back-propagation consists of "output grad" and "param grad", and that full-parameter back-propagation takes twice the computation time of the forward pass. These know results are not related to DP nor ZeRO and only summarized here for comparison with DP-ZeRO which additionally introduces the columns "communication" and "DP clip and noise". We meant by "figurative and dependent on settings" that the number 0.666 only gives a ballpark: it may be 0.555 for another model or dataset. The actual number is $\sum_{l\in \text{trainable layers}} 2BT_{l}^2(p_{l} + d_{l}) · I[2T_{l}^2< p_{l}d_{l}]$, which is not easy to explain in a few sentences, compared to the back-propagation or forward pass which only depends on $\Psi_{model}$. We meant by figurative that if all parameters are trainable, then DP-SGD takes about $2+2+2+0.666$ unit sof time compared to non-DP SGD's $2+2+2$ units of time, i.e. about 11\% slowdown; if only 1/1000 of parameters are trainable (as in LoRA), then DP-SGD takes $2+2+2/1000+0.666/1000$ units of time, i.e. almost no slowdown. This is roughly consistent with the empirical efficiency. We refer the reviewer to "Differentially Private Optimization on Large Model at Small Cost" Table 5 for theoretical complexity and Table 8 for empirical values. We will add it to the next revision!
>
> I wonder if the precision change can affect the precision of the noise and thus causing any problem / difference in privacy.
>
> **Response** This is a great question! Unfortunately this is under-investigated and no conclusion has been reached. The precision issue persists even for fp32 and fp64, or any precision formats because the computer only deals with *discrete* representation of numbers. For instance, the representation of $\pi$ is only accurate up to 7 digits in fp32 and up to 3 digits in fp16 (mixed precision training). In words, the mathematical gradient $g=activation*output\\_grad$ never matches the hardware representation exactly: $fp(activation) \cdot fp(output\\_grad)\neq fp(g)\neq g$ where fp is the precision format (fp16 or fp32 ...). In the DP regime, we will have
> $$\sum_i fp(C_i) fp(activation_i)\cdot fp(output\\_grad_i)+fp(noise)\neq fp(private\\_grad)\neq private\\_grad$$
>
> However, in practice, this precision issue is expected to be insignificant. Existing DP papers have observed successful defense over privacy attacks, even though fp32 is discrete. We believe the DP defense under fp16/bf16 will be similarly successful: consider the membership inference attack, which leverages the difference (e.g. in loss or gradient) between seen data and unseen data, e.g. seen data loss $\ll$ unseen data loss, regardless of the precision.

---

### Comment · Senior_Area_Chairs · 2023-11-14

Dear Authors,

Since this paper violates the formatting requirements, we unfortunately can't accept the paper to conference. For example, the paper uses much smaller left- and right-hand margins than what is required for a valid submission.

One possibility for the authors is to look into submission to a journal that allow for longer submissions.

Kind regards, Your SAC

---

> ### Author Response · Authors · 2023-11-14
>
> Sorry for the mis-formatting. We accidentally forgot to comment out \usepackage{fullpage} in latex. We have taken the initiative to update our revision to fulfill ICLR formatting requirements.
>
> Is this **remediation by re-formatting acceptable**, given that we are in the rebuttal phase? It would be a great pity to discontinue this discussion with the reviewers and disqualify the efforts we have put into this work. We sincerely appreciate your consideration.